# Rapid Analysis of Residues of 186 Pesticides in Hawk Tea Using Modified QuEChERS Coupled with Gas Chromatography Tandem Mass Spectrometry

**DOI:** 10.3390/ijerph191912639

**Published:** 2022-10-03

**Authors:** Xiao Shu, Nengming Chu, Xuemei Zhang, Xiaoxia Yang, Xia Meng, Junying Yang, Na Wang

**Affiliations:** 1Chongqing Academy of Agricultural Sciences, Chongqing 401329, China; 2Agricultural Product Quality and Safety Supervision, Inspection and Testing Center, Ministry of Agriculture and Rural Affairs, Chongqing 401329, China

**Keywords:** hawk tea, modified QuEChERS, GC-MS/MS, EMR-Lipid

## Abstract

In this work, the QuEChERS method was modified and evaluated for the determination of 186 pesticides from caffeine-free and fatty hawk tea prior to their gas chromatography tandem mass spectrometry analysis for the first time. The results showed that the combination of MgSO_4_ + PSA + MWCNTs plus EMR-Lipid provided the lowest matrix effect and best recovery; 117 of 186 pesticides manifested weak matrix effects. Thus, for accurate quantification, it is necessary to use matrix-matched calibration curves to compensate for the matrix effect. At the spiked level of 0.1 mg/kg, the average recoveries of 184 pesticides were in the range of 70–120% and the RSDs were 0.3–14.4% by the modified method. Good linearity was shown for 186 analytes at concentration of 0.01 mg/L~0.4 mg/L, and the correlation coefficients exceeded 0.99 for 182 pesticides. The detection limits of 186 pesticides by the modified QuEChERS method were 0.001–0.02 mg/kg, and the limits of quantification (LOQ) were 0.005 mg/kg~0.05 mg/kg. The necessity of solvent exchange is also explained in this work. The successful application of the modified QuEChERS in real samples proved that this method could be one of the routine options for analysis of herbal tea.

## 1. Introduction

Hawk tea, also named eagle tea, is widely known as an “affinal drug and diet” plant which means it was used as both medicine and food with hundreds of years of history in southern China. It is a caffeine-free but fatty tea which belongs to *Lauraceae* family and is different from the six main categories of teas (including green tea, white tea, yellow tea, oolong tea, black tea and dark tea). Hawk tea is considered a cooling and refreshing beverage to beat the heat by the rural people in southwest China owing to it being cheap and easily available. Spicy Bampa and Szechuan cuisine restaurants provide eagle tea as a complimentary drink to relieve customers after they have consumed spicy dishes. Therefore, its nutrient value and food safety are widely accepted by consumers. Previous studies [1] mainly focused on the active components of hawk tea, including mineral elements [2], volatile oils, [3] flavonoids, [4] and polysaccharides [5]. The content of fat in hawk tea is much higher than that in green tea, and its main components are listed in Appendix A [6]. Papers have demonstrated that hawk tea has the benefits of detoxification and anti-inflammatory, antioxidation, improvement of eyesight, and reduction of blood sugar and blood lipid etc., [7,8,9]. With the intensive and large-scale cultivation of eagle tea, the application of pesticides cannot be avoided, which causes a potential risk to public health. Many countries and regions have stipulated the maximum residue limits of pesticides in tea to regulate the use of pesticides. Pesticide levels detected in the mainstream teas commonly exceed the MRLs prescribed by the European Union (EU) and Codex Alimentarius Commission (CAC) [10]. Thus, it is essential to develop reliable, robust, and sensitive analytical methods to guarantee the safe consumption of hawk tea. 

The original version of the Quick, Easy, Cheap, Effective, Rugged, and Safe (QuEChERS) method was proposed by Anastassiades et al. in 2003 [11]. Modifications were made successively in the Association of Official Agricultural Chemists method with acetate buffer (AOAC 2007.01) [12] and the European Standard method (CEN 15662) with citrate buffer [13]. In recent years, the QuEChERS method has been further modified and widely used for multi-residue preparations in complex matrices because of its inherent properties: rapidity, simplicity, low cost, high efficiency, ruggedness, and safety [14,15]. At the same time, various materials have been investigated to enhance the clean-up capacity during the QuEChERS process, such as multiwalled carbon nanotubes (MWCNTs) and enhanced matrix removal (EMR) lipid. Multiwalled carbon nanotubes (MWCNTs) were first reported in 1991 [16], and have a unique nano-hollow structure, large specific surface area, and special chemical properties. Multiwalled carbon nanotubes have become one of the excellent solid phase extraction materials for the analysis of pesticide residues due to their strong adsorption capacity, stability, durability, and low cost [17,18]. EMR-Lipid is a novel material reported by the Agilent technology company in 2015. The composition of EMR-Lipid is undisclosed. It is used for removing lipids from QuEChERS extracts of fatty food matrix products, such as avocado and animal tissues, without loss of targeted compounds [19,20,21,22]. While hawk tea contains double the fat of green tea, EMR-Lipid may selectively remove it by the QuEChERS procedure. Gas chromatography tandem mass spectrometry (GC-MS/MS) is one of the most frequently used apparatus in analyzing pesticide residues, such as organochlorines and pyrethroids [23,24]. Research has suggested that GC-MS/MS is a reliable and robust analytical method with high sensitivity and selectivity [25,26]. It provides accurate qualitative and quantitative determination of hundreds of compounds in the presence of complex matrices. Besides, in terms of the apparatus cost, use cost, and accessibility, GC-MS/MS is superior to liquid chromatography-tandem mass spectrometry (LC-MS/MS). Over the past few years, most of the previous QuEChERS methods have focused on several kinds of pesticides in tea only [27,28]. There are few reports on the simultaneous determination of more than 100 pesticides combined being detected by the modified QuEChERS method with GC-MS/MS in tea, especially in herbal tea [29,30]. Hawk tea, caffeine-free but rich in fat, is a representative herbal tea that is distinguished from the six main kinds of teas, and accompanied with its rising consumption, there is a need to pay more attention to the food safety quality controls on it. However, there are few studies reporting the analytical technologies for detecting pesticide residues in hawk tea and the potential pesticide residue risks of hawk tea. 

In this work, fatty hawk tea was selected as the research object. The QuEChERS method was firstly modified by MWCNTs and with EMR-Lipid to evaluate their applicability for multi-pesticide residue determination in hawk tea combined with GC-MS/MS. Our analysis describes186 pesticide residues in hawk tea for the first time. The pre-treatment procedures and instruments conditions were optimized respectively. A series of parameters including recovery, precision, linearity, limit of detection (LOD), and limit of quantification (LOQ) were evaluated for the method validation. Finally, the proposed method was successfully applied to the determination of these pesticides in real hawk tea samples. 

## 2. Materials and Methods

### 2.1. Chemicals and Reagents

We purchased 186 pesticide reference material standards with purities greater than 95% from Sigma-Aldrich (Saint Louis, MO, USA) and Dr. Ehrenstorfer GmbH (Augsburg, Germany). MS-grade solvents, such as acetonitrile and ethyl acetate, were purchased from Merck Companies (Darmstadt, Germany); analytical grade acetic acid was obtained from Chengdu Kelong Chemical Reagent Company (Sichuan, China); analytical grade salts, such as sodium chloride (NaCl) and magnesium sulfate (MgSO_4_), were from Shanghai GuoYao Chemical Reagents (Shanghai, China). Graphitized Carbon Black (GCB, 40 μm), MWCNTs, and Primary Secondary Amine (PSA, 40 μm) were purchased from ANPEL Laboratory Technologies Inc. (Shanghai, China). EMR-Lipid was obtained from Agilent Technologies, Inc. (Santa Clara, CA, USA). Ultra-pure water was obtained using a MilliQ UF-Plus system (Millipore, Darmstadt, Germany) with a resistivity of at least 18.2 MΩ.cm at 25 °C.

### 2.2. Equipment and Experimental Conditions

Analysis was carried out using a Shimadzu GCMS-TQ8050 triple-quadrupole (QqQ) mass spectrometer with an electron ionization interface and an auto sampler AOC 20i/s. Chromatographic separation was achieved on a SH-Rxi-5Sil MS column (30 m × 0.25 mm i.d., 0.25 μm) from Shimadzu. 

Other equipment included: Vortex oscillator (IKA-Werke GmbH & CO. KG Janke & Kunkel, Staufen, Germany); centrifuge (Beckman Coulter, Inc., Indianapolis, IN, USA); nitrogen evaporator (Organomation Associates, Inc., Berlin, MA, USA); analytical balance with 0.0001 accuracy. Centrifuge tubes: 15 mL, 50 mL; pipettes: 1 mL, 5 mL, 10 mL; filter membrane: 0.22 μm.

### 2.3. QuEChERS Extraction Procedures

Two grams of each hawk tea powder was weighed into a 50 mL centrifuge tube containing EMR-Lipid materials. Then, 10 mL distilled water was added, shaken for 30 s, and left to stand for 30 min. Next, the samples were extracted with 10 mL acetonitrile, shaken for 30 s, put in 4 g NaCl and shaken by vortex for 1 min. Then, the samples were centrifuged at 5000 rpm for 10 min immediately. Afterwards, the upper 3 mL extraction solvent was transferred to a 15 mL QuEChERS purification falcon tube (contained with 1200 mgMgSO_4_, 400 mgPSA, 200 mg MWCNTs). Then, 3 mL toluene was accurately added to the tube. The tube was vortexed for 1 min and centrifuged for 10 min at 8000 rpm. The supernatant acetonitrile and toluene mixture layer (2 mL) was dried under N_2_ gas and re-dissolved with 1 mL ethyl acetate. It was passed through a nylon filter (0.22 μm) before analysis.

## 3. Results and Discussions

### 3.1. Optimization of GC/MS/MS Conditions

In this research, a total of 186 pesticides listed in Table 1 were selected for investigation, then one quantitation and at least two identification ions were adopted for scanning under the multi-reaction monitoring (MRM) model to avoid false positive response. First, in each different sample matrix, the same ions have a different response intensity, so the hawk tea matrix standard solution was used to establish a parent ion scan mode, with a scanning range from 50 to550 (m/z). Next, the precursor ions of the 186 pesticides were selected for the second collision of ions with different voltages. Then, we selected a signal value and the highest percentage collision value of the two ions and voltages. Finally, we determined the highest value for the quantitative ion, followed by the qualitative ion. On this occasion, pesticides could be successfully separated and identified via GC-MS/MS. For example, the retention times (RTs) of triadimefon and isazofos are 11.882 min and 11.886 min, respectively, and they cannot be separated and identified via gas chromatography. However, according to the differences of their quantitative ion pairs (208.10/111.00 * and 257.00/162.00 *) and qualitative ion pairs (208.10/127.00 and 257.00/119.00), under the multi-reaction monitoring (MRM) model of GC-MS/MS, these two pesticides were distinguished correspondingly. The RT of pesticides may vary slightly in different sample matrices. In order to avoid this, we added pesticides to the hawk tea matrix, and then changed the GC-MS/MS pressure to stabilize the retention time of the chromatography of the target compounds and ensure the retention times of the 186 pesticides were within the allowed range (±0.05 s). The quantitative ion collision voltage range for the 186 pesticides was 6–34 (eV) and the qualitative ion collision cover range was 6–30 (eV).

### 3.2. Optimization of QuEChERS Procedure

Although eagle tea is caffeine-free, it still contains other complex matrices such as pigments, carbohydrates, polyphenols, fat, and other substances. When dealing with complicated dry samples like hawk tea, selecting the appropriate pretreatment method and optimizing it during the sample preparation procedure can effectively reduce the amount of co-extracts and lower the matrix effect, at the same time keeping excellent accuracy. In our early-stage preparations, we found that with the QuEChERS original unbuffered method, the extraction procedure produced the fewest co-extracts and released the lowest heat compared with the acetate-buffered and citrate-buffered QuEChERS methods. Therefore, the original QuEChERS method was adopted and modified in this work. Then, affecting factors such as the absorbents and solvents were investigated successively.

#### 3.2.1. The Application of EMR-Lipid

The fat in hawk tea is twice that in green tea, so EMR-Lipid was considered to remove the fat. With high selectivity, EMR-Lipid could efficiently remove lipids based on volume exclusion and hydrophobic interaction mechanisms without loss of target analytes. Extraction and purification were included in the sample preparation procedure, so when applying EMR-Lipid, adding it to the extraction or the purification procedure should be taken into account. EMR-Lipid needs to activated by water and the hawk tea powder also needs to soak in water to improve the extraction efficiency. To simplify the steps, the hawk tea sample was weighed into a 50 mL tube containing the EMR-Lipid to remove the fat in the extraction step instead of in the purification step as a matter of priority; the water was then added to activate the EMR-Lipid materials at the same time to promote the dissolution of target components. The comparison experiment was also carried out. The EMR-Lipid was activated in the purification step; the purifying capacity was almost the same as when adding it in the extraction step, and so was the recovery result. Yet, when using it this way, there is one more purification and salting out procedure. As shown in Figure 1, it is obvious that the combination of MgSO_4_ + PSA + MWCNTs plus EMR-Lipid in the extraction procedure provided the better clean-up performance. After analysis of the hawk tea blank sample via the Q3scan mode, EMR-Lipid plus the combination of MgSO_4_ + PSA + MWCNTs showed the lower matrix effect (Appendix A). At the spike level of 0.1 mg/kg, the recovery rates of 167 of 186 pesticides (about 89.78%) were in the range of 70–120% (Appendix A).

#### 3.2.2. Selection of the Salt Composition

The different versions of QuEChERS have different salt composition; the original unbuffered QuEChERS adopted the combination of 4 g MgSO_4_ +1 g NaCl. Magnesium sulfate will release a lot of heat when dissolved in water, which will accelerate the dissolution of all substance and then lead to a stronger matrix effect (which was also why AOAC 2007.01 and CEN15662 QuEChERS were excluded in this work). The comparison experiment proceeded by adding 4 g NaCl in the extraction procedure. Gravimetric analysis and temperature monitoring were conducted for the two kinds of salt composition. The average amounts of five replicates were evaluated for each salt composition (Table 2). The average weight of 5 mL co-extracts in the combination of 4 g MgSO_4_ +1 g NaCl and 4 g NaCl was about 0.13103 g and 0.10930 g, respectively. In addition, the thermometer showed that the highest value in the extraction process can rise from room temperature to 40.5 °C using the combination of 4 g MgSO_4_ +1 g NaCl. However, the temperature was almost unchanged when used 4 g NaCl. MgSO_4_ was exothermic when dissolved in water whereas the NaCl was not; a relatively high dose of MgSO_4_ may have not only caused the degradation of some pesticides but also promoted the dissolution of other substances and exacerbated the matrix effect, so the weight of the co-extracts was heavier with the salt combination of 4 g MgSO_4_ + 1 g NaCl. It goes without saying that single NaCl was better for sample extraction.

#### 3.2.3. Pretreatment Effects of Different Combinations of Adsorbents 

The original unbuffered QuEChERS method involved extraction with acetonitrile and purification with a certain quantity of adsorbents, including PSA and/or C18 and/or GCB. Acetonitrile turned out to be an optimal solvent in analyzing pesticide residues with relatively fewer total extracts [31]. For the sample purification procedure, PSA forms hydrogen bonds through amine groups and polar matrix components, which is mainly in favor of removing organic acids, polar pigments, fatty acids, and sugars in the sample; [32] MgSO_4_ is mainly used to absorb water and ensure the adsorption capacity of PSA. GCB has a strong adsorption capacity and can effectively remove chlorophyll in tea [33]. However, in recent years, the MWCNTs have been proven to perform better when combined with PSA or some other materials in QuEChERS. The GCB was replaced by MWCNTs in the purification procedure in some research [18]. C18 was not adopted in the purification procedure, because EMR-Lipid was used in the extraction procedure to remove the fat. Based on this situation, three combinations of adsorbents were investigated for the purification procedure: MgSO_4_ + PSA, MgSO_4_ + PSA + GCB, MgSO_4_ + PSA + MWCNTs. In Figure 1, it can be visually observed that the combination of MgSO_4_ + PSA + MWCNTs in the extraction procedure provided preferable clean-up performance. Besides, the recoveries also suggested that the combination of MgSO_4_ + PSA + MWCNTs was better: at the spike level of 0.1 mg/kg, 162 of 186 (87.10%) pesticides were in the range of 70–120% while the GCB combination gave the result that 158 of 186 (84.94%) pesticides were in the range of 70–120% (Appendix A). Although both GCB and MWCNTs had the characteristics of adsorbing planar structure pesticides, such as quinoxyfen, ditalimfos, and cyprodinil, MWCNTs reduced the adsorbability to a certain extent for some of the pesticides, such as aclonifen and boscalid, according to our results. When GCB was used, the recovery of some pesticides was lower, compared with the same dose of MWCNTs. The reason may be that the multi-ring planar structure of GCB has a certain adsorption effect on planar structure pesticides, leading to a low recovery rate of planar structure pesticides, while the hollow structure of MWCNTs has less adsorption effect on these pesticides [34]. Therefore, the combination of MgSO_4_ + PSA + MWCNTs was selected as the absorbent in this experiment.

#### 3.2.4. The Addition of Toluene

Even though MWCNTs are an efficient sorbent in removing complex matrices and provided good clean-up performance, there still remains the stubborn problem that MWCNTs may absorb certain pesticides with a planar structure and reduce their recovery. For further improving the recovery of these target pesticides that possessed planar structure, toluene was added after the upper extractant was transferred to the tube for purification. The benzene ring from toluene may compete with MWCNTs and reduce the tendency of pesticides to be absorbed by MWCNTs [17]. In this way, the recovery showed breakthrough improvement: 184 of 186 pesticides were in the range of 70–120%, which was a proportion of 98.92%. The exceptions were phosfolan and quinoxyfen. These two pesticides not only have a planar structure but also have weak solubility in water, especially phosfolan, so their recovery rates were 59.0% and 65.4%, as shown in Table 3.

#### 3.2.5. The Exchange of Solvent

Despite the fact that the recovery of the pesticides was increased dramatically by adding toluene, it was still essential to exchange the solvent before analysis by GC-MS/MS, especially for the targets with RT less than 14 min. We found that when using the mixture of acetonitrile and toluene as the solvent directly determined by GC-MS/MS, this often caused a dilemma that was ignored in some previous work [17]. Although the accuracy was guaranteed by toluene, with high molecular polarity, tailing of peaks was more likely to occur when using acetonitrile as the solvent. In addition, due to the solvent effect caused by mixing the solvents acetonitrile and toluene, the chromatographic peaks broadened and bifurcated, retention time drifted, and double peaks happened more frequently, which would significantly influence the qualitative and quantitative results. Ethyl acetate was a splendid choice to resolve this contradictory situation, as it is less volatile than n-hexane and more environmentally friendly than toluene. The comparison of some typical pesticides verified by the two solvents is shown in Appendix A. The chromatographic peaks immediately became symmetrical and smooth and better qualitative and quantitative results were obtained after exchange by ethyl acetate. For long-term monitoring, the exchange of solvent also has crucial benefits for instrument life and maintenance.

### 3.3. Matrix Effects Study

The QuEChERS methods are widely used in vegetables, fruit, and teas, and the matrix effect is greatly influenced by the materials applied in modification and optimization processes [35,36]. Matrix effects were assessed by comparing the slopes of six matrix-matched calibration curves to the slopes of the calibration curves in solvent. Matrix effects were calculated with Equation [37]: ME (%)$=\left(\frac{\mathrm{slope}\;\mathrm{of}\;\mathrm{calibration}\;\mathrm{curve}\;\mathrm{in}\;\mathrm{matrix}}{\mathrm{slope}\;\mathrm{of}\;\mathrm{calibration}\;\mathrm{curve}\;\mathrm{in}\;\mathrm{solvent}}-1\right)*100$. Calibration curves (6 points from 0.01 to 0.4 mg/L) were plotted by solvent (ethyl acetate) and matrix (blank hawk tea solutions obtained from the preparation procedure by modified QuEChERS). When the ME% is within −20 and +20 it is considered a low matrix effect and if ME% is within −50~−20 and +20~+50 it is considered a medium matrix effect. If more than −50 or +50, the matrix effect was evaluated as a strong matrix effect. The MEs (%) for the modified QuEChERS method are depicted in Figure 2. It was obvious that there existed enhanced matrix effects by GC-MS/MS overall in our study. However, compared to standard solution peaks, matrix effects also possess the merits of improving peak shapes with less tailing, more symmetry, and higher intensity [29]. Even though the matrix effect will always exist when applying GC-MS/MS, the modified QuEChERS with the combination of MgSO_4_ + PSA + MWCNTs plus EMR-Lipid succeeded in minimizing the matrix effect to the weakest level compared with other work [38]: 117 of 186 pesticides (62.9%) had weak matrix effects, 45 pesticides (24.2%) had moderate matrix effects, and 24 pesticides (12.9%) had strong matrix effects. In contrast, the matrix effect of the QuEChERS without EMR-lipid and MWCNTs was the stronger: 62 pesticides (33.3%) showed weak matrix effects, 69 pesticides (37.1%) had moderate matrix effects, and 55 pesticides (29.6%) had strong matrix effects. In order to visually observe which modified QuEChERS method provided better cleaning of the extracts, gas chromatography tandem mass spectrometry (GC-MS/MS) analysis of the extracts in full scan mode was carried out as a complementary evaluation (Appendix A). The chromatogram verified the above matrix effect calculation results.

### 3.4. Recoveries and RSDs

The 186 pesticides of mixed standard solution were added to pesticide-free hawk tea powders, and the recovery rates and variabilities of the modified QuEChERS were evaluated at concentrations of 0.02 mg/kg, 0.05 mg/kg, and 0.1 mg/kg with three replicates. The results are presented in Table 3 and Appendix A. By comparing reproducibility and recovery rates, it was found that the QuEChERS modified by the MWCNTs and EMR-Lipid improved the recovery of some of the pesticides. Moreover, it was apparent that the addition of toluene distinctively raised the recoveries of planar structure pesticides, keeping the recoveries at an ideal range of 70–120% for 184 pesticides at the concentration of 0.1 mg/kg. Relative standard deviations (RSDs, %) were less than 14.4%. Comparing these results with the published SPE method, the modified QuEChERS combined with GC-MS/MS performed much better in efficiency, recovery, and repeatability [39] in comparison with some other modified extraction methods: the extraction efficiency and general suitability all improved, meanwhile the matrix effect was lower [40,41]. These excellent results showed that the QuEChERS has irreplaceable superiority, with wide applicability, stability, accessibility, and simplicity. Even dicofol, which was said to perform better in QuEChERS acetate, showed a decent recovery at the concentration of 0.1 mg/kg by this method [42,43], which indicated that the QuEChERS method had preeminent accuracy for quantification of multi-pesticide residues in hawk tea. At the concentration of 0.02 mg/kg, the recoveries of 93.01% pesticides were in the range of 60–130% and the RSDs of 180 pesticides were less than 15. At the concentration of 0.05 mg/kg for the modified QuEChERS, the recoveries of 86.02% pesticides were in the range of 70–120% and the RSDs of 182 pesticides were below 15.

### 3.5. Linearity

Working standard solutions of 0.01, 0.025, 0.05, 0.1, 0.2, and 0.4 mg/L were prepared by the modified QuEChERS method to obtain the matrix-matched linearity. Linear ranges and correlation coefficients are summarized in Table 3. It is conspicuous that the correlation coefficients for the tested 182 pesticide residues were all higher than 0.99, and those of propazine, desmetryn, iprodione, and difenoconazole were more than 0.98. The calibration curves were linear within the range.

### 3.6. Limits of Detection and Limits of Quantitation

Limits of detections (LODs) and limits of quantifications (LOQs) were determined by adding different concentrations of pesticide standards to the hawk tea blank samples. When the signal-to-noise ratio reached three for each pesticide, the corresponding concentration was regarded as the LOD of the method. Meanwhile, when the signal-to-noise ratio reached 10 for each pesticide, the corresponding concentration was fixed as the limit of quantification (LOQ) of the method [44]. The LODs and LOQs of the 186 pesticides are listed in Table 3. For the 186 pesticides, the LODs and LOQs ranged from 0.001 to 0.02 mg/kg and 0.005 to 0.05 mg/kg, respectively. The results indicated that analyzing the pesticide residues by GC/MS/MS in hawk tea produced good quantitative detection limit findings. In addition, the LOQs for the analysis compounds were lower than the maximum residue limits (MRLs) specified by China and the Codex Alimentarius Commission (CAC). Therefore, the established GC/MS/MS analysis method is worth using for routine analysis of pesticide residues in hawk tea or other kinds of herbal tea.

### 3.7. Real Samples

Twenty-six samples of eagle tea were purchased from the major hawk tea producing areas in Wulong and Wushan District, Chongqing, China to verify the method. Among all the tested analytes, chlorpyrifos and bifenthrin were detected with a concentration of 0.0054 mg/kg and 0.0106 mg/kg in two samples from Wulong District, which meant eagle tea in this area has a potential risk of pesticide contamination. The international Codex Alimentarius Commission (CAC) standards and China national standards GB2763-2021 provided that the MRLs of chlorpyrifos poisoning in tea were 0.1 mg/kg and 2 mg/kg, and the MRLs of bifenthrin were 5 mg/kg and 30 mg/kg, respectively. The obtained results indicated that the abuse of some pesticides is unavoidable, although the residue levels of pesticides were lower than MRLs established by different countries and organizations. Regular monitoring of hawk tea samples for multi-residue pesticides is still necessary and important, and a tighter management and regulation of pesticides needs to be implemented in hawk tea production and marketing to guarantee the safety of the tea drinking public.

## 4. Conclusions

In this paper, the QuEChERS method was modified using EMR-Lipid and MWCNTs to lower the matrix effect and improve recovery, and then applied in the determination of 186 pesticides in hawk tea combined with GC–MS/MS for the first time. The addition of toluene was the crucial part for improving the recovery of planar structure pesticides. Moreover, the exchange of solvent also played an important role in wiping out the solvent effect and improved the chromatographic peak for better qualitative and quantitative analysis purposes. The recoveries for all pesticides were excellent; even dicofol recovery worked well with the modified QuEChERS although the literature has reported that acetate-buffered QuEChERS is more fit for these kinds of pesticides. The calibration parameters for the modified QuEChERS method including recovery, precision, linear range, LOD, and LOQ were examined, which indicated that modified QuEChERS coupled with GC–MS/MS was suitable for rapid multi-pesticides analysis in hawk tea. Analysis of real samples revealed that the abuse of pesticides still exists. Therefore, regular and long-term monitoring of pesticide residues in hawk tea and herbal tea is of great significance.

## Figures and Tables

**Figure 1 ijerph-19-12639-f001:**
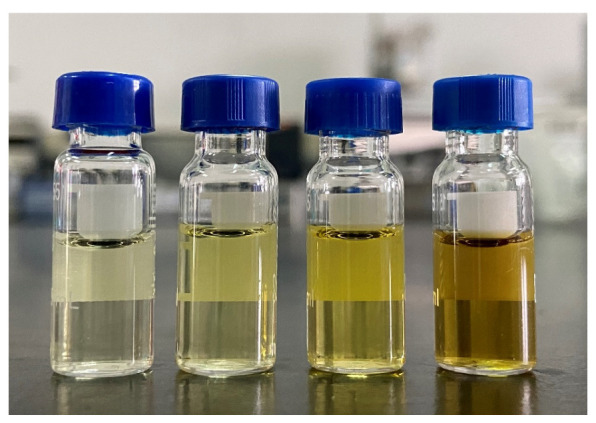
Comparison of color of different sorbents in hawk tea samples: from the right to the left are MgSO_4_ + PSA, MgSO_4_ + PSA +GCB, MgSO_4_ + PSA +MWCNTs, MgSO_4_ + PSA +MWCNTs + EMR-Lipid, respectively.

**Figure 2 ijerph-19-12639-f002:**
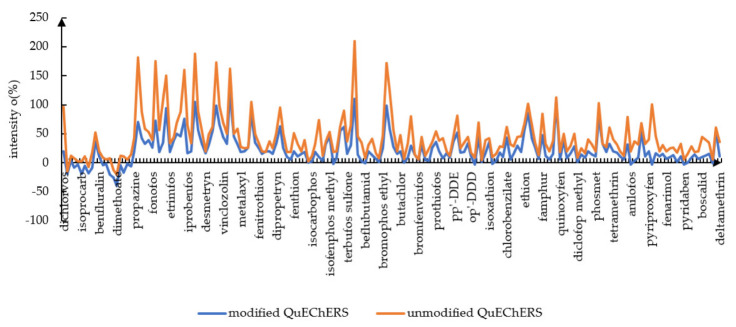
The comparison of matrix effect by modified QuEChERS and unmodified QuEChERS.

**Table 1 ijerph-19-12639-t001:** GC-MS/MS parameters of 186 pesticides in selected reaction monitoring (MRM) mode.

No.	Pesticides	RT/min	Qualitative Ion Pairs (*m*/*z*)	CE/	No.	Pesticides	RT/min	Qualitative Ion Pairs (*m*/*z*)	CE/
				eV					eV
1	dichlorvos	5.972	109.00 > 79.00 *	8	94	op’-DDE	14.811	246.00 > 176.00 *	30
			185.00 > 93.00	14				248.00 > 176.00	28
2	dichlorobenzonitrile	6.935	170.90 > 136.00 *	14	95	paclobutrazol	14.893	236.10 > 125.00 *	14
			170.90 > 100.00	24				236.10 > 167.00	10
3	biphenyl	7.389	154.10 > 128.10 *	22	96	butachlor	14.92	176.10 > 147.10*	14
			154.10 > 115.10	24				188.10 > 160.10	12
4	etridiazole	8.034	210.90 > 182.90 *	10	97	fenothiocarb	14.947	160.10 > 72.00 *	10
			182.90 > 139.90	18				160.10 > 106.10	12
5	propoxur	8.181	152.00/110.10 *	18	98	ditalimfos	15.046	130.00 > 102.10 *	10
			110.00/82.00	8				148.00 > 130.10	10
6	isoprocarb	8.946	136.00 > 121.00 *	10	99	butamifos	15.065	286.10 > 202.10 *	14
			121.00 > 77.00	22				200.10 > 65.00	22
7	tecnazene	9.553	260.90 > 202.90 *	14	100	napropamide	15.155	128.00 > 72.10 *	6
			202.90 > 142.90	22				100.00 > 72.00	8
8	diphenylamine	9.941	169.10 > 66.00 *	24	101	bromfenvinfos	15.164	267.00 > 159.00 *	15
			167.10 > 139.10	28				269.00 > 161.00	21
9	ethoprophos	10.016	200.00 > 158.00 *	6	102	fluorodifen	15.182	190.00 > 126.00 *	12
			158.00 > 97.00	18				190.00 > 75.00	21
10	chlorpropham	10.263	127.10 > 65.00 *	22	103	flutolanil	15.227	173.00 > 145.00 *	14
			213.10 > 171.10	6				173.00 > 95.00	26
11	benfluralin	10.382	292.10 > 264.00 *	8	104	chlorfenson	15.236	175.00 > 111.00 *	12
			292.10 > 160.00	22				175.00 > 75.00	28
12	sulfotep	10.414	322.00 > 202.00 *	10	105	hexaconazole	15.263	214.00 > 159.00 *	20
			322.00 > 174.00	18				214.00 > 172.00	20
13	monocrotophos	10.532	127.10 > 109.00 *	12	106	prothiofos	15.272	266.90 > 238.90 *	10
			127.10 > 95.00	16				309.00 > 238.90	14
14	phorate	10.629	260.00 > 75.00 *	8	107	fludioxonil	15.29	248.00 > 127.00 *	26
			231.00 > 129.00	24				248.00 > 154.00	20
15	alpha BHC	10.735	180.90 > 144.90 *	16	108	pretilachlor	15.318	262.10 > 202.10 *	10
			218.90 > 182.90	8				238.10 > 162.10	10
16	dimethoate	11.008	125.00 > 47.00 *	14	109	isoprothiolane	15.318	231.10 > 189.00 *	10
			125.00 > 79.00	8				290.10 > 118.00	14
17	simazine	11.166	201.10 > 173.10 *	6	110	profenofos	15.363	338.90 > 268.90 *	18
			201.10 > 186.10	6				336.90 > 266.90	14
18	atrazine	11.25	215.10 > 58.00 *	14	111	pp’-DDE	15.435	246.00 > 176.00 *	30
			215.10 > 173.10	6				317.90 > 248.00	24
19	beta BHC	11.261	180.90 > 144.90 *	16	112	oxadiazon	15.453	258.00 > 175.00 *	8
			218.90 > 182.90	8				302.00 > 175.00	14
20	clomazone	11.292	204.10 > 107.00 *	20	113	DEF	15.489	202.00 > 147.00 *	6
			204.10 > 78.00	26				202.00 > 113.00	20
21	propazine	11.313	229.10 > 58.00 *	14	114	dieldrin	15.525	276.90 > 241.00 *	8
			229.10 > 187.10	6				262.90 > 193.00	34
22	gamma-BHC	11.408	180.90 > 144.90 *	16	115	myclobutanil	15.534	179.10 > 125.00 *	14
			218.90 > 182.90	8				179.10 > 152.00	8
23	profluralin	11.429	318.00 > 199.10 *	18	116	op’-DDD	15.562	235.00 > 165.00 *	24
			318.00 > 55.10	18				237.00 > 165.00	28
24	terbuthylazine	11.492	229.10 > 173.10 *	6	117	oxyfluorfen	15.571	252.00 > 196.00 *	22
			214.10 > 71.00	16				361.00 > 300.00	14
25	terbufos	11.502	231.00 > 128.90 *	26	118	bupirimate	15.579	273.10 > 108.10 *	16
			231.00 > 174.90	14				273.10 > 193.10	8
26	fonofos	11.576	137.10 > 109.10 *	8	119	kresoxim methyl	15.605	116.00 > 89.00 *	15
			246.00 > 137.10	6				116.00 > 63.00	30
27	pronamide	11.586	172.90 > 144.90 *	16	120	cyflufenamid	15.761	118.10 > 90.00*	10
			172.90 > 109.00	26				118.10 > 89.00	25
28	diazinon	11.639	304.10 > 179.10 *	10	121	isoxathion	15.813	177.10 > 130.10 *	10
			179.10 > 137.10	18				177.10 > 116.10	12
29	pyrimethanil	11.723	198.10 > 183.10 *	14	122	cyproconazole 1	15.822	139.10 > 111.10 *	16
			198.10 > 118.10	28				222.10 > 125.10	24
30	isazofos	11.886	257.00 > 162.00 *	8	123	fluazifop butyl	15.9	282.00 > 91.10 *	18
			257.00 > 119.00	18				282.00 > 238.10	18
31	etrimfos	11.967	181.10 > 153.10 *	10	124	nitrofen	15.909	202.00 > 139.00 *	24
			292.10 > 181.10	8				282.90 > 253.00	12
32	delta-BHC	11.987	180.90 > 144.90 *	16	125	endrin	15.917	262.90 > 191.00 *	30
			218.90 > 182.90	8				244.90 > 173.00	32
33	triallate	11.998	268.10 > 184.00 *	20	126	chlorobenzilate	16.064	139.00 > 111.00 *	16
			270.10 > 186.00	20				251.00 > 139.00	14
34	tebupirimfos	12.099	261.10 > 137.10 *	18	127	fensulfothion	16.108	293.00 > 125.00 *	14
			318.10 > 152.10	14				293.00 > 153.00	8
35	pirimicarb	12.13	238.10 > 166.10 *	12	128	diniconazole	16.151	268.00 > 232.00 *	12
			166.10 > 55.00	20				270.00 > 234.00	10
36	iprobenfos	12.15	204.00 > 91.00 *	8	129	oxadixyl	16.212	163.10 > 132.10 *	8
			204.00 > 122.00	12				132.10 > 117.10	18
37	formothion	12.282	170.00 > 93.00 *	8	130	pp’-DDD	16.229	235.00 > 165.00 *	24
			224.00 > 125.00	18				237.00 > 165.00	28
38	pentachloroaniline	12.303	262.80 > 191.90 *	21	131	ethion	16.229	153.00 > 97.00 *	14
			264.80 > 193.90	21				230.90 > 129.00	24
39	phosphamidon	12.425	127.10 > 109.10 *	12	132	op’-DDT	16.264	235.00 > 165.00 *	24
			127.10 > 95.10	18				237.00 > 165.00	28
40	dichlofenthion	12.445	279.00 > 222.90 *	14	133	chlorthiophos	16.264	324.90 > 268.90 *	14
			222.90 > 204.90	14				268.90 > 205.00	18
41	desmetryn	12.475	213.00 > 171.10 *	6	134	aclonifen	16.307	212.00 > 182.10 *	15
			213.00 > 58.10	18				264.00 > 194.10	18
42	propanil	12.516	217.00 > 161.00 *	10	135	triazophos	16.521	161.00 > 134.00 *	8
			160.90 > 99.00	24				161.00 > 106.00	14
43	acetochlor	12.547	174.10 > 146.10 *	12	136	famphur	16.646	218.00 > 109.00 *	16
			223.10 > 132.10	22				218.00 > 79.00	24
44	phenthoate	12.58	273.9/125.0 *	20	137	benalaxyl	16.671	148.10 > 105.10 *	16
			273.9/246.0	6				148.10 > 79.10	24
45	malaoxon	12.618	126.90 > 99.00 *	10	138	carbophenothion	16.729	157.00 > 45.00*	18
			268.00 > 126.90	10				341.90 > 157.00	14
46	vinclozolin	12.658	212.00 > 172.00 *	16	139	trifloxystrobi	16.762	116.00 > 89.00 *	15
			285.00 > 212.00	12				131.00 > 89.00	30
47	parathion methyl	12.709	263.00 > 109.00 *	14	140	edifenphos	16.779	173.00 > 109.00 *	10
			125.00 > 47.00	12				310.00 > 173.00	14
48	tolclofos methyl	12.719	264.90 > 249.90 *	14	141	propiconazole	16.829	173.00 > 145.00 *	16
			264.90 > 93.00	24				259.00 > 69.00	14
49	alachlor	12.73	188.10 > 160.10 *	10	142	quinoxyfen	16.912	237.10 > 208.10 *	28
			188.10 > 132.10	18				307.10 > 237.10	22
50	ametryn	12.877	227.10 > 185.10 *	6	143	pp’-DDT	16.929	235.00 > 165.00 *	24
			227.10 > 58.00	14				237.00 > 165.00	28
51	metalaxyl	12.877	249.20 > 190.10 *	8	144	hexazinone	17.046	171.10 > 71.00 *	16
			206.10 > 132.10	20				171.10 > 85.00	16
52	ronnel	12.906	284.90 > 269.90 *	16	145	tebuconazole	17.196	250.10 > 125.10 *	22
			286.90 > 271.90	18				125.10 > 89.00	18
53	prometryn	12.936	226.10 > 184.10 *	10	146	diclofop methyl	17.212	340.00 > 253.00 *	14
			241.20 > 184.10	12				253.00 > 162.00	22
54	pirimiphos methyl	13.141	290.10 > 125.00 *	22	147	piperonylbutoxide	17.334	176.10 > 131.10 *	12
			290.10 > 233.10	12				176.10 > 117.10	20
55	terbutryn	13.18	241.20 > 185.10 *	6	148	epoxiconazol	17.454	192.00 > 138.00 *	14
			241.20 > 170.10	14				192.00 > 111.00	26
56	fenitrothion	13.2	277.00 > 260.00 *	6	149	pyridaphenthion	17.654	340.00 > 199.10 *	8
			277.00 > 109.10	14				199.10 > 92.00	16
57	ethofumesate	13.239	207.10 > 161.10 *	8	150	iprodione	17.678	187.00 > 124.00 *	25
			207.10 > 137.10	12				243.90 > 187.00	5
58	bromacil	13.307	204.90 > 187.90 *	14	151	phosmet	17.798	160.00 > 77.00 *	24
			206.90 > 189.90	16				160.00 > 133.00	14
59	phorate sulfoxide	13.366	153.00 > 97.00 *	12	152	bifenthrin	17.822	181.10 > 166.10 *	12
			199.00 > 171.10	6				181.10 > 179.10	12
60	malathion	13.376	173.10 > 99.00 *	14	153	EPN	17.846	156.90 > 77.00 *	24
			173.10 > 127.00	6				169.10 > 77.00	22
61	dipropetryn	13.444	255.00 > 222.20 *	9	154	bromopropylate	17.869	340.90 > 182.90 *	18
			255.00 > 180.20	18				340.90 > 184.90	20
62	metolachlor	13.464	162.10 > 133.10 *	16	155	piperophos	17.877	320.10 > 122.10 *	14
			238.10 > 162.10	12				140.10 > 98.00	12
63	phoratesulfone	13.493	153.00 > 97.00 *	12	156	tetramethrin	17.893	164.10 > 107.10 *	14
			153.00 > 125.00	6				164.10 > 77.00	22
64	chlorpyrifos	13.503	196.90 > 168.90 *	14	157	methoxychlor	17.965	227.10 > 169.10 *	24
			313.90 > 257.90	14				227.10 > 212.10	14
65	thiobencarb	13.532	100.00 > 72.00 *	5	158	etoxazole	17.973	141.00 > 113.00*	15
			125.00 > 89.00	18				141.00 > 63.10	30
66	fenthion	13.591	278.00 > 109.00 *	20	159	fenamidone	18.037	238.00 > 237.20 *	10
			278.00 > 169.00	14				268.10 > 180.10	16
67	parathion	13.659	109.00 > 91.00 *	6	160	tebufenpyrad	18.1	333.10 > 171.10 *	20
			148.90 > 119.00	5				333.10 > 276.10	8
68	isofenphos oxon	13.689	229.10 > 201.00 *	10	161	anilofos	18.131	226.10 > 157.00 *	14
			201.00 > 121.00	20				226.10 > 184.00	6
69	triadimefon	13.718	208.10 > 181.00 *	10	162	bifenox	18.162	340.90 > 309.90 *	10
			208.10 > 111.00	22				340.90 > 188.90	20
70	buprofezin	13.726	175.10/132.10 *	14	163	tetradifon	18.354	226.90 > 199.00 *	16
			175.10/117.10	12				355.90 > 159.00	18
71	isocarbophos	13.737	289.10 > 136.00 *	14	164	phosalone	18.462	182.00 > 111.00*	14
			230.00 > 212.00	10				182.00 > 138.00	8
72	dicofol	13.803	139.00 > 111.00 *	16	165	leptopho	18.47	376.90 > 361.90 *	24
			139.00 > 75.00	28				374.90 > 359.90	24
73	trichloronat	13.84	297.00 > 269.00 *	15	166	pyriproxyfen	18.631	136.10 > 78.00 *	20
			299.00 > 271.00	15				136.10 > 96.00	14
74	pirimiphos ethyl	13.915	304.00 > 168.00 *	10	167	iambda cyhalothrin	18.631	208.00 > 181.00 *	8
			318.00 > 166.00	15				197.00 > 141.00	12
75	bromophos	13.925	330.90 > 315.90 *	14	168	mefenacet	18.708	192.00 > 136.00 *	14
			328.90 > 313.90	18				192.00 > 109.00	24
76	isofenphos methyl	14.019	199.00 > 121.00 *	14	169	acrinathrin	18.77	289.10 > 93.00 *	14
			241.10 > 121.10	22				289.10 > 77.00	26
77	fosthiazate	14.019	195.00 > 103.00 *	10	170	pyrazophos	18.981	221.10 > 193.10 *	12
			195.00 > 60.00	22				221.10 > 149.10	14
78	pendimethalin	14.141	252.10 > 162.10 *	10	171	fenarimol	19.003	251.00 > 139.00 *	14
			252.10 > 191.10	8				330.00 > 139.00	8
79	chlorfenvinphos	14.15	323.00 > 267.00 *	16	172	azinphos ethyl	19.13	160.10 > 132.10 *	4
			267.00 > 159.00	18				132.10 > 77.00	14
80	cyprodinil	14.169	224.10 > 208.10 *	16	173	permethrin 1	19.598	183.10 > 153.10 *	14
			224.10 > 197.10	22				183.10 > 168.10	14
81	terbufos sulfone	14.235	153.00 > 97.00 *	21	174	coumaphos	19.712	362.00 > 109.00 *	16
			199.00 > 97.00	21				362.00 > 226.00	14
82	fipronil	14.244	366.90 > 212.90 *	30	175	fluquinconazole	19.734	340.00 > 298.00 *	20
			368.90 > 214.90	30				340.00 > 313.00	14
83	penconazole	14.272	248.10 > 157.10 *	26	176	pyridaben	19.756	147.10 > 117.10 *	22
			159.10 > 123.10	22				147.10 > 132.10	14
84	phosfolan	14.301	255.00 > 227.00 *	6	177	dioxathion	19.77	152.90 > 96.90 *	10
			255.00 > 140.00	22				185.00 > 129.00	12
85	isofenphos	14.338	213.00 > 121.00 *	15	178	fenbuconazole	20.121	198.10 > 129.10 *	10
			213.00 > 185.00	6				129.10 > 102.10	18
86	beflubutamid	14.46	176.00 > 91.10 *	15	179	cyfluthrin	20.136	226.10 > 206.10 *	14
			221.00 > 193.00	12				198.90 > 170.10	25
87	quinalphos	14.47	146.10 > 118.00 *	10	180	cypermethri	20.46	163.10 > 127.10 *	6
			146.10 > 91.00	24				163.10 > 91.00	14
88	mephosfolan	14.498	196.00 > 140.00 *	12	181	boscalid	20.522	140.10 > 112.10 *	12
			196.00 > 168.00	6				140.10 > 76.00	24
89	procymidone	14.535	283.00 > 96.00 *	10	182	flucythrinate	20.626	199.10 > 157.10 *	10
			285.00 > 96.00	10				157.10 > 107.10	12
90	triadimenol	14.545	168.10 > 70.00 *	10	183	fenvalerate	21.338	225.10 > 119.10 *	20
			128.10 > 65.00	22				225.10 > 147.10	10
91	bromophos ethyl	14.721	358.90 > 302.90 *	16	184	fluvalinate	21.452	250.10 > 55.00 *	20
			302.90 > 284.90	18				250.10 > 200.00	20
92	methidathion	14.739	145.00 > 85.00 *	8	185	difenoconazole	21.793	323.00 > 265.00 *	14
			145.00 > 58.00	14				265.00 > 202.00	20
93	chlordane trans	14.757	374.80 > 265.90 *	26	186	deltamethrin	22.109	180.90 > 151.90 *	22
			372.80 > 263.90	28				252.90 > 93.00	20

Note: “*” in the table represents quantitative ion pairs. RT represents retention time. CE represents collision energy.

**Table 2 ijerph-19-12639-t002:** Weight of 5 mL co-extracts in the two salts’ composition.

No.	4 g NaCl	4 g MgSO_4_ + 1 g NaCl
	Co-Extracts (g)	Co-Extracts (g)
replicate 1	0.10805	0.12837
replicate 2	0.10925	0.13428
replicate 3	0.11052	0.13125
replicate 4	0.10892	0.13172
replicate 5	0.10977	0.12953
average	0.10930	0.13103

**Table 3 ijerph-19-12639-t003:** Recoveries, relative standard deviations (RSDs), limits of detections (LODs), and limits of quantifications (LOQs) of 186 pesticides using the modified QuEChERS.

No.	Pesticides	R^2^	Spiked 0.02 mg/kg		Spiked 0.05 mg/kg		Spiked 0.1 mg/kg		LODs mg/kg	LOQs mg/kg
			Recovery (%)	RSD	Recovery (%)	RSD	Recovery (%)	RSD		
1	dichlorvos	0.9985	81.9	6.1	88.7	5.2	106.3	4.2	0.01	0.02
2	dichlorobenzonitrile	0.9901	64.8	11.6	78.6	3	94	5.7	0.01	0.02
3	biphenyl	0.9906	100.3	21.5	87	5	89.3	3.5	0.005	0.01
4	etridiazole	0.9973	118.7	3.3	87.6	1.8	101.2	7.2	0.005	0.01
5	propoxur	0.9965	56.2	5.2	68.8	8.4	73.6	8.4	0.02	0.05
6	isoprocarb	0.9976	85	6.4	71.8	9.4	96	6.7	0.005	0.01
7	tecnazene	0.9942	61.9	6.4	64.9	11	104.9	11.3	0.01	0.02
8	diphenylamine	0.9962	103.8	15.5	95.6	10.5	103.5	5.9	0.005	0.01
9	ethoprophos	0.9981	97.8	12.6	98.7	16.7	96.1	14.4	0.005	0.01
10	chlorpropham	0.9976	66.6	4.8	70.4	13.1	81.6	5.4	0.01	0.02
11	benfluralin	0.9992	107.7	16.6	95.5	5.3	111.9	5.6	0.01	0.02
12	sulfotep	0.999	85.7	20.2	81.2	21.3	92.3	6.8	0.01	0.02
13	monocrotophos	0.9993	76.1	12.2	70.2	4.2	110.4	2.1	0.005	0.01
14	phorate	0.9896	56.5	10.3	62.2	4.2	81.4	0.9	0.02	0.05
15	alpha-BHC	0.9986	79.1	9.3	85.4	13.1	104.1	9.1	0.005	0.01
16	dimethoate	0.9922	82.1	20.8	79.4	6.5	94.6	8.1	0.005	0.01
17	simazine	0.9984	60.7	6.1	71.4	9.4	84.2	4.5	0.01	0.02
18	atrazine	0.9907	85.4	12.8	77.8	5.7	91.4	1.2	0.01	0.02
19	beta-BHC	0.9987	90.8	13.5	88.5	8.2	108.8	2.1	0.005	0.01
20	clomazone	0.9992	61.9	4.3	59.8	9.7	97	7.5	0.01	0.02
21	propazine	0.9946	96.9	6.7	72	14.4	95.4	13	0.01	0.02
22	gamma- BHC	0.9991	106.6	3	105.3	8.8	119.4	2.3	0.005	0.01
23	profluralin	0.9985	54.1	2	49.4	14.7	93.5	2.4	0.01	0.02
24	terbuthylazine	0.9532	50	26	48.7	16	91.2	5.5	0.02	0.05
25	terbufos	0.998	115.4	7.2	133.8	6.1	116.2	8.1	0.01	0.02
26	fonofos	0.9986	101.7	9.9	101.3	10.9	105.9	10.5	0.01	0.02
27	pronamide	0.9967	104.8	6.1	94.7	10.3	103.8	1.8	0.005	0.01
28	diazinon	0.9935	88.3	3.7	80.9	6.7	105.1	1.9	0.005	0.01
29	pyrimethanil	0.9955	54.9	7.4	56.7	11	76	6.5	0.02	0.05
30	isazofos	0.9991	58.8	7.8	65	11.4	104.3	7.4	0.01	0.02
31	etrimfos	0.9963	93.2	3	100.3	2.5	114.7	6.6	0.005	0.01
32	delta- BHC	0.9921	76	6.9	78.4	5.5	105.6	7.9	0.005	0.01
33	triallate	0.9974	81.5	14.5	110.2	6.2	99.1	9.7	0.005	0.01
34	tebupirimfos	0.9952	82.2	8.6	83.7	5.1	92.9	4.6	0.01	0.02
35	pirimicarb	0.9998	85.4	8.4	82.6	8.7	92.7	9.1	0.005	0.01
36	iprobenfos	0.9962	114.3	8.8	99.1	9.1	103.5	5.8	0.005	0.01
37	formothion	0.9989	70.2	5.2	72.3	2.2	78	1	0.005	0.01
38	pentachloroaniline	0.9979	94.4	11.1	89.3	5.1	100.3	1.1	0.005	0.01
39	phosphamidon	0.998	102.3	7.3	86.5	6.9	96.5	5.2	0.005	0.01
40	dichlofenthion	0.997	66.2	3.9	70.9	10.8	103	2.3	0.01	0.02
41	desmetryn	0.9851	65.7	9	65.5	3	80.9	4.4	0.01	0.02
42	propanil	0.9995	98.5	14.4	84.6	1.7	94	10.4	0.005	0.01
43	acetochlor	0.9923	76	10.7	90.6	5.1	111.7	7	0.01	0.02
44	phenthoate	0.9959	65.3	5.6	70.3	2.1	75.5	1.9	0.005	0.01
45	malaoxon	0.9992	62.5	11.1	70.9	14.5	70.6	1.2	0.01	0.02
46	vinclozolin	0.99	76.6	3.3	78.4	14.8	97.5	9	0.01	0.02
47	parathion methyl	0.9989	70.8	4.4	76.6	11.3	85.4	8.9	0.005	0.01
48	tolclofos methyl	0.9971	74.8	6.4	90.8	2.4	104.1	2.6	0.01	0.02
49	alachlor	0.9989	117.1	3.4	108.8	10.3	118.9	5.2	0.005	0.01
50	ametryn	0.9921	64.2	14.8	68.5	6.3	82.5	3.6	0.01	0.02
51	metalaxyl	0.9987	81.3	13.1	80.6	6.1	86.1	7	0.005	0.01
52	ronnel	0.9994	65.8	11.9	65.7	3.8	75.5	0.3	0.01	0.02
53	prometryn	0.9908	66.1	7.5	71.1	11.4	78.4	1.4	0.01	0.02
54	pirimiphos methyl	0.9927	67.2	6.6	82	8.7	100	2.3	0.01	0.02
55	terbutryn	0.9975	79.4	6	82.4	4.2	97.9	4.2	0.005	0.01
56	fenitrothion	0.9959	91.1	6.4	99.6	0.1	118.5	5.8	0.005	0.01
57	ethofumesate	0.9962	95.5	6.5	100.5	4.3	107.4	3.6	0.005	0.01
58	bromacil	0.9989	77	8.2	73.6	10.7	88.6	4.6	0.005	0.01
59	phorate sulfoxide	0.9964	78	7	93.3	4.9	117	5.6	0.01	0.02
60	malathion	0.9969	88.7	5.9	95.3	1.6	111.5	2	0.005	0.01
61	dipropetryn	0.9902	72	6.6	76.8	4.9	91.3	2.9	0.005	0.01
62	metolachlor	0.9921	82.5	6	85.9	1.3	97.3	2.4	0.005	0.01
63	phoratesulfone	0.9971	50	9	41.1	7.5	96.4	3.8	0.02	0.05
64	chlorpyrifos	0.9989	94.7	5.3	88.1	10.5	98.7	11.2	0.005	0.01
65	thiobencarb	0.9921	70.7	5.7	69.9	10.3	95.3	4.2	0.01	0.02
66	fenthion	0.9987	73	3.8	90.6	3.1	94.2	3.7	0.005	0.01
67	parathion	0.9914	126.5	5.3	104.5	2.1	114.2	2	0.005	0.01
68	isofenphos oxon	0.9988	86.9	3.5	84.8	1.4	100.1	2.7	0.005	0.01
69	triadimefon	0.9987	77.8	4	92.5	5.2	107.9	7.6	0.005	0.01
70	buprofezin	0.9979	65.2	1.3	70.8	3.3	73.2	8.7	0.01	0.02
71	isocarbophos	0.9959	91	3	87.8	10.5	99.1	3.2	0.005	0.01
72	dicofol	0.9962	76.9	9.2	73.5	6	93.3	1.4	0.005	0.01
73	trichloronat	0.9991	77.2	9.2	86.2	4.1	93.2	2.7	0.005	0.01
74	pirimiphos ethyl	0.9963	59.5	6.2	67.4	2	75.4	2.7	0.02	0.05
75	bromophos	0.9921	84.9	5.7	88.3	4.3	97.1	0.7	0.01	0.02
76	isofenphos methyl	0.9974	74.7	5.3	87.6	2.1	99.9	0.8	0.005	0.01
77	fosthiazate	0.9952	68.7	13.2	67.8	4.7	95	2.9	0.01	0.02
78	pendimethalin	0.9992	76.4	9.5	91.5	2.3	102.5	4.9	0.005	0.01
79	chlorfenvinphos	0.9962	79.2	2.9	86	2.4	102.7	3.1	0.005	0.01
80	cyprodinil	0.9989	61.8	6.6	65.4	7.3	79.7	2.9	0.01	0.02
81	terbufos sulfone	0.9979	84.1	1.8	93.7	8.3	107.8	2.9	0.005	0.01
82	fipronil	0.9911	75.1	9.7	80.9	2.7	100.8	5.2	0.005	0.01
83	penconazole	0.9963	65.2	4.8	76.8	10.5	88.2	3.5	0.01	0.02
84	phosfolan	0.9921	53.4	11.2	45.7	14.2	58.6	6.4	0.02	0.05
85	isofenphos	0.9974	88.9	4.1	90.7	5.5	101.4	2.3	0.005	0.01
86	beflubutamid	0.9952	80.2	5.2	91.9	1.9	100.7	0.5	0.005	0.01
87	quinalphos	0.9905	81.8	3.3	92.7	2.2	87.2	1.4	0.005	0.01
88	mephosfolan	0.9962	67.7	4.5	71.3	6.8	79.5	4.1	0.01	0.02
89	procymidone	0.9989	82.8	4.3	89.5	7.5	99	5	0.005	0.01
90	triadimenol	0.9979	122.8	2.8	116.4	11.7	98.4	3.8	0.01	0.02
91	bromophos ethyl	0.9901	88.8	4.5	95.2	4.3	103.6	2.6	0.005	0.01
92	methidathion	0.9963	89.2	9.2	96.2	1.7	106.3	0.3	0.005	0.01
93	chlordane trans	0.9921	79.8	0.5	86.2	14	90.4	2.9	0.01	0.02
94	op’-DDE	0.9974	73	4.3	78.2	1.6	85.7	0.6	0.005	0.01
95	paclobutrazol	0.9952	90.8	6.2	94.5	0.9	110.1	2.7	0.005	0.01
96	butachlor	0.9908	112.9	2.4	106	2.2	108.3	0.9	0.005	0.01
97	fenothiocarb	0.9962	81.3	3.1	80.9	2.7	90.9	1.2	0.005	0.01
98	ditalimfos	0.9989	57.8	4.5	59.9	5.1	79.8	1.7	0.01	0.02
99	butamifos	0.9979	73	5.8	75.7	2.9	86.3	2.2	0.01	0.02
100	napropamide	0.9901	82.3	3.5	89.9	4.7	96.1	5.2	0.01	0.02
101	bromfenvinfos	0.9963	87.7	5.6	87.3	3.1	97.4	0.5	0.01	0.02
102	fluorodifen	0.9921	104.8	4.7	108.5	6.1	123.6	4.8	0.005	0.01
103	flutolanil	0.9974	87.6	3.4	86.1	1.8	98.3	0.4	0.005	0.01
104	chlorfenson	0.9952	95.9	7.2	90.3	2.7	102	0.4	0.01	0.02
105	hexaconazole	0.9988	79.6	5.3	86.9	8.6	98.4	7.5	0.01	0.02
106	prothiofos	0.9962	57.8	5.9	64.5	1.8	70.6	2.8	0.01	0.02
107	fludioxonil	0.9989	78.9	1.8	89.2	7.8	97.3	1.6	0.01	0.02
108	pretilachlor	0.9979	78.6	6.7	86.6	1.7	98.9	5	0.005	0.01
109	isoprothiolane	0.9991	68.9	11.7	74.1	4.8	83.2	3	0.01	0.02
110	profenofos	0.9963	76	5.3	83.6	1.6	95	2.8	0.005	0.01
111	pp’-DDE	0.9921	68.4	12.7	72.7	2.6	83.3	2.1	0.005	0.01
112	oxadiazon	0.9974	73.9	7.1	83.6	5.6	88.8	4.2	0.005	0.01
113	DEF	0.9952	75.6	1.3	85	6.5	92.2	3.3	0.005	0.01
114	dieldrin	0.9908	66.9	3.9	67.5	7	72.8	2.7	0.005	0.01
115	myclobutanil	0.9962	86.3	6	88.1	3.2	102.9	1.6	0.01	0.02
116	op’-DDD	0.9989	69.7	7.3	78.2	3.9	90.1	1.3	0.005	0.01
117	oxyfluorfen	0.9979	119.3	5.4	106.7	10.1	119.7	1.6	0.005	0.01
118	bupirimate	0.9911	52.2	4.3	52.9	1.3	71.4	1.2	0.02	0.05
119	kresoxim methyl	0.9963	92.1	7.9	89.9	3.7	97.2	1.2	0.005	0.01
120	cyflufenamid	0.9921	82.6	3.8	83.9	8.4	92.1	4.6	0.005	0.01
121	isoxathion	0.9974	79.8	3.8	95.4	7.5	102.9	5.4	0.01	0.02
122	cyproconazole 1	0.9952	88.1	6.1	73.2	22.3	99.5	3.2	0.01	0.02
123	fluazifop butyl	0.9922	92.5	5.9	86.4	3.4	104.3	3.2	0.01	0.02
124	nitrofen	0.9962	86.1	9.1	86.1	5.7	98.7	5.3	0.005	0.01
125	endrin	0.9989	77.8	2.8	83.4	5.8	116.2	11.1	0.01	0.02
126	chlorobenzilate	0.9979	103.4	3.8	99.2	2.2	95.9	2.2	0.005	0.01
127	fensulfothion	0.9991	86.1	2.3	97.3	5.4	103.3	5.7	0.01	0.025
128	diniconazole	0.9963	71.2	2.7	70.5	1.3	87.8	2.7	0.01	0.02
129	oxadixyl	0.9921	93	4.6	91.1	1.5	102.7	2.5	0.01	0.02
130	pp’-DDD	0.9974	71.9	2.9	77.7	3.1	90.5	1.5	0.005	0.01
131	ethion	0.9952	86.3	3.5	94.5	3.2	109.9	2	0.005	0.01
132	op’-DDT	0.9998	73.9	4.2	77.7	3.1	90.5	1.5	0.005	0.01
133	chlorthiophos	0.9962	90.8	2.9	88	7.6	98	0.5	0.005	0.01
134	aclonifen	0.9989	108.3	5.5	94.5	8.8	103.4	7.2	0.005	0.01
135	triazophos	0.9979	122.5	5.5	101.5	3.5	108.7	3	0.005	0.01
136	famphur	0.9991	68.9	2.5	74	3	78.9	0.8	0.01	0.02
137	benalaxyl	0.992 3	81	6.5	91.1	1.5	104.1	0.6	0.005	0.01
138	carbophenothion	0.992 1	90.8	8.9	88.1	2.2	100.9	2.6	0.005	0.01
139	trifloxystrobi	0.997 4	85.6	4.8	86.4	2.9	99.5	2.2	0.005	0.01
140	edifenphos	0.995 2	84.8	0.6	86.9	3.9	101.5	1.2	0.005	0.01
141	quinoxyfen	0.9991	52.1	13.3	60.5	3.8	65.1	3.1	0.02	0.05
142	propiconazole	0.9962	85.5	4.1	88.2	10.4	97.9	7	0.005	0.01
143	pp’-DDT	0.9989	66.7	5.7	78.5	0.9	85.2	1.1	0.01	0.02
144	hexazinone	0.9929	66.7	5.3	72.1	2.2	79	1.1	0.01	0.02
145	tebuconazole	0.9991	83.4	3.4	85.7	3.2	92.6	3.9	0.005	0.01
146	diclofop methyl	0.9963	82.1	3.9	90.3	5.3	100.1	0.4	0.005	0.01
147	piperonylbutoxide	0.9921	71.6	5.2	81.4	1	89.7	1.3	0.01	0.02
148	epoxiconazol	0.9904	75.6	4.9	78.5	3.4	90.7	4.7	0.01	0.02
149	pyridaphenthion	0.9952	68.8	1.9	70.9	5.5	77.1	1	0.01	0.02
150	iprodione	0.9808	123	6.7	113.1	7.7	106.8	3.4	0.005	0.01
151	phosmet	0.9962	69.2	8.4	68.3	1.8	76.4	2.9	0.01	0.02
152	bifenthrin	0.9902	63.4	4.1	77.2	0.4	87	1.6	0.001	0.005
153	EPN	0.997	99.2	10.2	97.6	6	107.5	3.6	0.005	0.01
154	bromopropylate	0.9921	70.6	4.1	81.9	3.1	93.7	0.4	0.01	0.02
155	piperophos	0.9913	79.7	1.2	82.8	2	93.6	2.1	0.01	0.02
156	tetramethrin	0.9921	74.4	5.4	82.1	1.2	92.1	0.9	0.01	0.02
157	methoxychlor	0.9904	71.7	5.4	86.3	4.3	95.5	1.4	0.01	0.02
158	etoxazole	0.9952	76.7	8.5	95.4	10.4	102.6	1.7	0.01	0.02
159	fenamidone	0.9908	79.5	5.3	88.4	2.7	99	1.6	0.01	0.02
160	tebufenpyrad	0.9962	81.7	9.2	84	2.8	90.5	0.7	0.005	0.01
161	anilofos	0.9989	80.5	6.5	93.3	3.4	102.8	3.7	0.005	0.01
162	bifenox	0.9979	80.3	1.2	90.4	10.5	104	3	0.005	0.01
163	tetradifon	0.9901	78.7	6.6	82.5	3.2	93.3	0.5	0.01	0.02
164	phosalone	0.9963	63.3	8.3	67.7	2.3	78.7	1.9	0.01	0.02
165	leptopho	0.9921	61	6.5	65.8	3.9	74.9	1.4	0.01	0.02
166	pyriproxyfen	0.9974	80.5	5.8	82.5	1.6	91.3	1.9	0.005	0.01
167	iambda cyhalothrin	0.9952	81.3	6.3	87.9	6.2	98.1	3.5	0.005	0.01
168	mefenacet	0.9998	83	3.8	88.3	1	98.4	0.8	0.005	0.01
169	acrinathrin	0.9962	73	8.3	80.7	2	93.6	4	0.01	0.02
170	pyrazophos	0.9982	76.8	5.3	73.1	1.4	79.8	2	0.01	0.02
171	fenarimol	0.9979	73.7	7.2	80.3	3.4	94	6.5	0.01	0.02
172	azinphos ethyl	0.9951	84.3	9	84.2	7.9	80.9	1.7	0.01	0.02
173	permethrin	0.9963	85.5	7.9	93.1	7.4	89.1	6.1	0.005	0.01
174	coumaphos	0.9921	80.2	8.1	89.3	2.1	100.7	2.3	0.005	0.01
175	fluquinconazole	0.9904	87.4	7.3	91.4	4.5	94.9	2.1	0.005	0.01
176	pyridaben	0.9992	73.7	7.3	76.1	1.7	87	1.5	0.01	0.02
177	dioxathion	0.9998	72.3	7.1	78	9.5	91.7	5.3	0.01	0.02
178	fenbuconazole	0.9952	81.2	4.8	89.1	1.2	103.5	4.5	0.005	0.01
179	cyfluthrin	0.9989	74.7	7	82	7.2	96.4	2	0.01	0.02
180	cypermethri	0.991	76.3	7.8	78.4	8.2	89.1	1.2	0.01	0.02
181	boscalid	0.9921	69.4	1.2	76.7	1.4	83.5	1	0.01	0.02
182	flucythrinate	0.9974	76.7	4.4	80.4	1.3	94.9	0.7	0.01	0.02
183	fenvalerate	0.9902	74.5	4.7	75.5	2.1	94.8	1.1	0.01	0.02
184	fluvalinate	0.9926	84.4	9.6	83	2.9	96.2	2.2	0.005	0.01
185	difenoconazole	0.9822	92.1	2.6	100.9	5	105.7	4.8	0.005	0.01
186	deltamethrin	0.9902	78.8	4.1	87.5	6.2	88.8	2.2	0.01	0.02

## Data Availability

All data are shown in the main manuscript.

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
