# Peer review of "Rapid Analysis of Residues of 186 Pesticides in Hawk Tea Using Modified QuEChERS Coupled with Gas Chromatography Tandem Mass Spectrometry"

_ijerph, 2022, doi:10.3390/ijerph191912639_

Round 1
Reviewer 1 Report
The manuscript entitled "Rapid analysis of 186 pesticide residues in hawk tea using modified QuEChERS coupled with gas chromatography tandem mass spectrometry" is an interesting study, supported by an impressive amount of data. The experiment was well-designed and performed thoroughly. The authors modified the QuEChERS method by the EMR-lipid and MWCNTs, and successfully applied in determination of 186 pesticides in hawk tea with GC–MS/MS. It is quite an improvement to the previously known methods.
The paper is written in a concise way, and the results are presented as clearly as it is possible with so many results. Although there is no part of the manuscript that would not be understandable, language definitely should be improved before publication.
After corrections, I recommend publication of this manuscript.
Some minor issues are:
Figure 2 seems to be missing (?)
Table 2 was it 3 g or 4 g of NaCl ? - there is inconsistency between the text and the table
latin names should be written in italics (beginning of introduction)
A loose thought: determination of pesticides in tea powder surely is important but also challenging as it causes a great deal of problems during sample preparation. But, from the perspective of safety for tea consumers, wouldn't it be enough to determine pesticides in tea extracts, prepared in a similar way that this kind of tea is usually served?
Reviewer 2 Report
Comments:
C1 - Line 363
The "splendid" term should be replaced with "excellent" or " acceptable".
C2 - paragraph 3.7 real samples
The authors should report the anlytical results in mg/kg unit - the same of the MRL in order to support the reading process of work.
C3 - paragraph 3.6 limit od detection and limits of quantitation
The Limits LOD, LOQ should be reported in the same unit of the analytical results preferably in mg/kg.
